# Microstructure and Mechanical Properties of Spark Plasma Sintered Mg-Zn-Ca-Pr Alloy

**Bartłomiej Hrapkowicz** [1,*], **Sabina Lesz** [1,*], **Małgorzata Karolus** [2], **Dariusz Garbiec** [3], **Jakub Wiśniewski** [3], **Rafał Rubach** [3], **Klaudiusz Gołombek** [4], **Marek Kremzer** [5] and **Julia Popis** [1]

1   Department of Engineering Materials and Biomaterials, Silesian University of Technology, 18a Konarskiego St, 44-100 Gliwice, Poland; julia.popis@polsl.pl
2   Institute of Materials Engineering, University of Silesia, 1a 75 Pulku Piechoty St, 41-500 Chorzow, Poland; karolus@us.edu.pl
3   Łukasiewicz Research Network — Poznań Institute of Technology, 6 Ewarysta Estkowskiego St, 61-755 Poznan, Poland; dariusz.garbiec@pit.lukasiewicz.gov.pl (D.G.); jakub.wisniewski@pit.lukasiewicz.gov.pl (J.W.); rafal.rubach@pit.lukasiewicz.gov.pl (R.R.)
4   Materials Research Laboratory, Silesian University of Technology, 18a Konarskiego St, 44-100 Gliwice, Poland; klaudiusz.golombek@polsl.pl
5   Nanotechnology and Materials Technology Scientific and Didactic Laboratory, Silesian University of Technology, 7a Towarowa St, 44-100 Gliwice, Poland; marek.kremzer@polsl.pl
*   Correspondence: bartlomiej.hrapkowicz@polsl.pl (B.H.); sabina.lesz@polsl.pl (S.L.)

**Abstract:** Alloys based on magnesium are of considerable scientific interest as they have very attractive mechanical and biological properties that could be used to manufacture biodegradable materials for medical applications. Mechanical alloying is a very suitable process to obtain alloys that are normally hard to produce as it allows for solid-state diffusion via highly energetic milling, producing fine powders. Powders obtained by this method can be sintered into nearly net-shape products, moreover, their phase and chemical composition can be specifically tailored. This work aims to investigate the effect of milling time on the density, microstructure, phase composition, and mechanical properties of Mg-Zn-Ca-Pr powders processed by high energy mechanical alloying (HEMA) and consolidated by spark plasma sintering (SPS). Thus, the results of XRD phase analysis, particle size distribution (granulometry), density, mechanical properties, SEM investigation of mechanically alloyed and sintered Mg-Zn-Ca-Pr alloy are presented in this manuscript. The obtained results illustrate how mechanical alloying can be used to produce amorphous and crystalline materials, which can be sintered and demonstrates how the milling time impacts their microstructure, phase composition, and resulting mechanical properties.

**Keywords:** metallic alloys; Mg-based alloy; high energy mechanical alloying; spark plasma sintering

## 1. Introduction

Materials engineering is concerned with understanding the relationship between a material's composition, microstructure, and properties and the influence of manufacturing processes on a material's microstructure. In materials engineering, the focus is on how to translate or transform materials into useful devices and structures. Material selection is the act of choosing the material best suited to achieve the requirements of a given application. Many different factors go into determining the selection requirements, such as mechanical properties, chemical properties, physical properties, electrical properties, and cost [1–3].

The materials used nowadays for craniofacial and orthopedic applications have one major flaw. Although they are inert, being permanent fixtures, they can perform well, but after playing their role, for example, stabilizing a bone, they remain in the patient's body. The various plates, pins, and screws, which are used to secure the fixture, are usually deemed necessary to remove. The process of removal is a second invasive surgical procedure, which causes more stress and discomfort to the treated patient. Due to this fact,

it is necessary to develop novel materials capable of biodegradation, meaning controllable and more importantly predictable corrosion along with the resorption of the implant constituents. Developing such materials would help tremendously and create a new branch of materials in a field mainly dominated by biodegradable polymeric materials [4–6].

Magnesium and its alloys have always attracted significant interest as potential materials for medical applications due to their properties such as comparable mechanical properties to human bone tissue, good biocompatibility, biodegradability, lightweight, and appropriate corrosion behavior for this kind of application [6–12]. The mechanical properties and the elastic modulus of magnesium are especially important as the more they are comparable to the natural human bone, the lower the possibility of inducing stress shielding in a bone [12,13]. While the abovementioned properties are critical, corrosion resistance is one of the most crucial traits, as it is necessary to finely control and predict it to employ a biodegradable material. The low corrosion resistance of pure magnesium is a flaw in conventional engineering applications, but for biomedical applications it is an advantage. The products of magnesium degradation are soluble, non-toxic hydroxides metabolized by the organism and excreted in the urine.

Unfortunately, during magnesium dissolution, large amounts of hydrogen are released, which may cause severe damage to surrounding tissues, as they are not accepted by the human body and may cause the formation of hydrogen pockets and inflammations [6,12,14]. As such the dissolution rate and the corrosion characteristic of such a magnesium-based implant need to be finely tailored in order to reach an acceptable level to be applicable. Those characteristics are affected by factors such as the general alloy composition and its microstructure. Thankfully, these can be adjusted via processing parameters and techniques [15,16].

As reported by several researchers, amorphous magnesium alloys exhibit high strength, ductility, and the possibility of controlled degradation, as opposed to their crystalline counterparts [6,17–19]. The metallic glass bears additional advantages over the crystalline, traditional alloys as it lacks the grain boundaries, which improves its corrosion resistance. Moreover, they possess greater mechanical strength and hardness [20,21]. Zberg et al. [22,23] carried out in vivo studies confirming the biocompatibility of the magnesium alloys and the possibility of reducing hydrogen production by zinc addition. Those claims were further proven by Wang et al. [19,24] and Gu et al.'s [19,24] in vitro studies. However, it is worth noting that the amorphous materials are extremely difficult to prepare. They require very specific conditions to be able to solidify rapidly, where the critical thickness is usually very limited [25,26].

High energy mechanical alloying (HEMA), however, entirely disregards the necessity of complex melting procedures, which usually need advanced studies in order to understand the processes of melting and solidification [27]. On the other hand, HEMA is a low-energy solid-state processing method that is capable of synthesis of the amorphous phases in powder form. They can be further processed into a bulk form or near-net shape. The most important advantage of the HEMA process is the ability to make the issues with maximum critical thickness obsolete [10,28–30]. The powder produced by the method is refined in both size and structure. It is possible to choose the milling parameters to a very finely optimized degree, allowing for grain refinement and control over the crystallinity or amorphous structures of the milled material. The high degree of freedom over designing such a material would result in the ability to produce materials with desired properties in powdered form, hence they can be consolidated with methods of additive manufacturing or sintering, such as park plasma sintering (SPS) [31–33]. Spark plasma sintering is an advanced powder metallurgy technique that can be used in the production of implants from Mg alloys. SPS allows fast densification at lower temperature in contrast to conventional sintering techniques, resulting in the retention of nanocrystalline structures and non-equilibrium alloys produced by HEMA, and associated properties [34,35]. SPS is advantageous for sintering metallic powders due to its ability to produce a sample with an ultra-fine microstructure without defects and pores [36,37]. This process also helps

in retaining the refined grain size, which is obtained after significant refinement through milling [38]. The grain growth can be minimized by controlling the consolidation parameters, e.g., holding time and temperature, stress applied to the sample during sintering. Recent work attempts to improve the consolidation of nanocrystal-line Mg-based alloy powder using SPS, thereby aiming towards better mechanical and corrosion properties [39].

Therefore, there is merit in optimizing the SPS process parameters to produce fully dense nanocrystalline alloys with extended solid solubility. The union between those innovative methods for preparing nanocrystalline and amorphous materials (mechanical alloying and spark plasma sintering as a consolidation process), allow one to obtain Mg-alloys which have values of strength, density and porosity which enable them to be considered as materials for clinical applications [40,41].

The aim of this work is to investigate the effect of milling time on the density, microstructure, phase composition, and mechanical properties of Mg-Zn-Ca-Pr powders processed by HEMA and consolidated by SPS. Thus, the results of the XRD phase analysis, particle size distribution (granulometry), density, mechanical properties, SEM investigation of mechanical alloyed and sintered Mg-Zn-Ca-Pr alloy are presented in this manuscript.

## 2. Materials and Methods

The alloy powder with a nominal atomic composition of $Mg_{65}Zn_{30}Ca_4Pr_1$ was prepared via the HEMA method. The powder mixture for the process was prepared with Mg, Zn, Pr powders with a purity of 99.99%, and Ca pieces with a purity of 99.99% closed in a tightly sealed stainless steel container in a high purity (99.99%) argon atmosphere. The 8000D Mixer/Mill—Dual High Energy Ball Mill (SPEX SamplePrep, Metuchen, NJ, USA) was used for the mechanical synthesis. The vials were milled for various cycles consisting of 1 h milling and 30 min cooldown breaks. The ball-to-powder ratio was set to 10:1, as the optimal ratio reported by various studies [30,42–44]. The balls were made of 316L stainless steel and were 10 mm in diameter. The samples varied in milling time, the milling times were as follows: 8, 13, 20, 30, and 70 h.

The X-ray diffraction measurements of the obtained alloys were performed using the Empyrean diffractometer (PANalytical, Almelo, The Netherlands) with Cu-K $\alpha$ radiation and PIXCell counter, employing the step-scanning method in 10 to 150° 2θ angle range. The phase analysis of substrates and milling products was performed with the High Score Plus PANalytical software (version 4.0, PANalytical, Almelo, The Netherlands) and the ICDD PDF4+ 2016 database (International Centre for Diffraction Data, Newtown Square, PA, USA). The structural characteristic determination of the unit cell parameters, crystallite sizes and lattice strains of the observed phases were performed using the Rietveld refinement [45,46] and Williamson–Hall theory [47–49] implemented in the High Score Plus software.

The particle size distribution of the powders was assessed with Analyssette 22 MicroTec+ (Fritsch, Weimar, Germany) in ethyl alcohol.

The spark plasma sintering of powders was performed in an HP D 25/3 device (FCT Systeme, Rauenstein, Germany) in an argon atmosphere with a sintering temperature of 350 °C and a compaction pressure of 50 MPa, with a holding time of 4 min and a heating rate of 50 °C·min$^{-1}$ up to 300 °C and 25 °C·min$^{-1}$ from 300 °C to 350 °C. Sintering was carried out using 2334-grade graphite tools (MERSEN, Gennevilliers, France) with a diameter of 20 mm. To enhance the conductivity of the contacts and to prevent sticking, powder was separated from the tool elements by Papyex N998 graphite foil (MERSEN, Gennevilliers, France). Sintering curves of the samples milled for 13, 20, and 70 h are presented in Figure 1a–c, respectively. The punch displacement curve reflects the densification process of the powder, as well as its stabilization after reaching the sintering temperature [50].

Density measurements were performed with the helium pycnometer AccuPyc II 1340 (Micromeritics, Norcross, GA, USA) on the powder mixture and sintered samples. The principle of determining the volume of a given sample of material is based on the assumption that it is the part of the void chamber that was not occupied by the introduced

gas. Helium is a gas used to "flush" samples to accurately determine their volume. The sample's mass is inserted into the device after weighing it on a precise analytical scale [51].

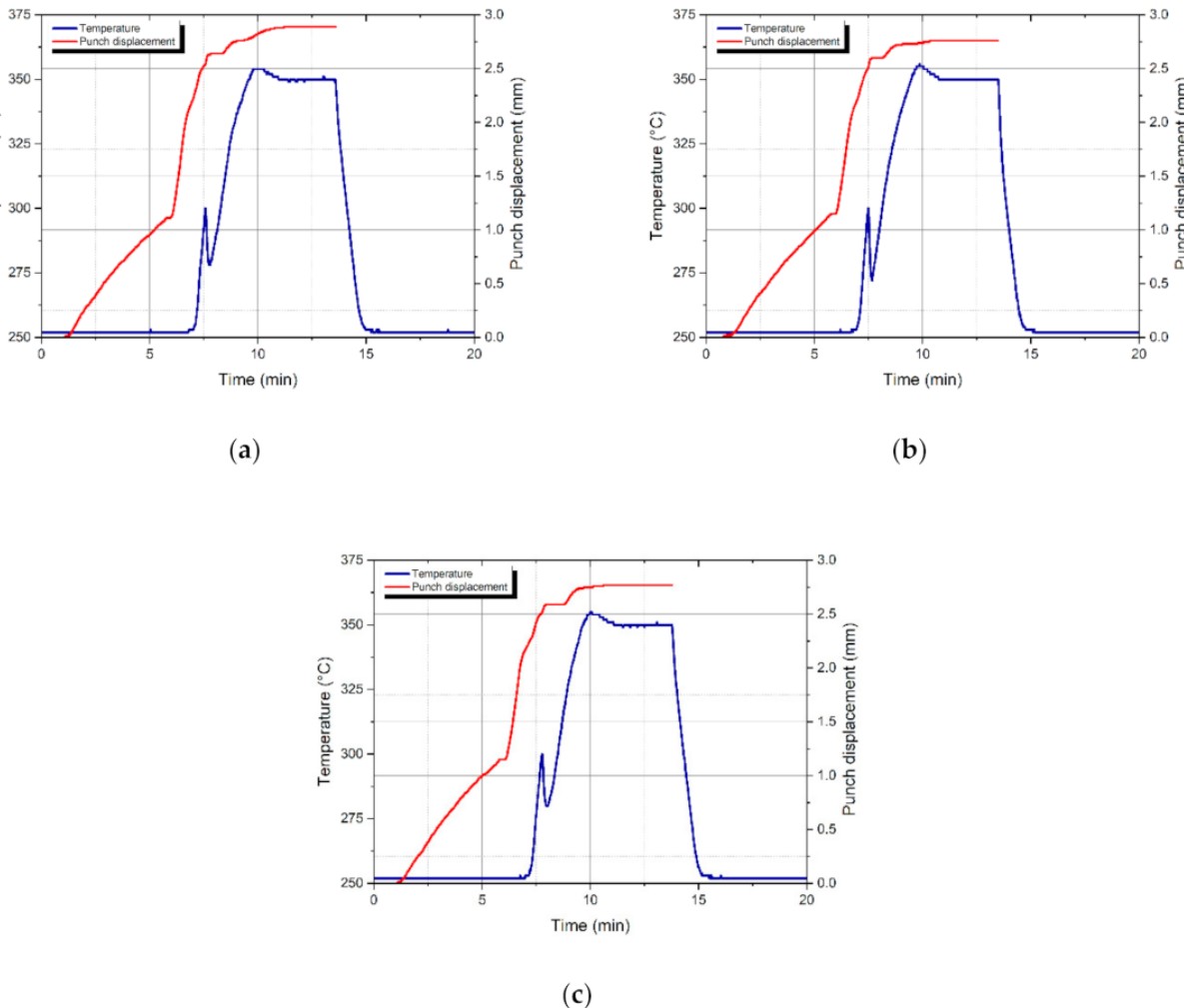

**Figure 1.** Temperature change and punch displacement recorded during the SPS of powders milled for (**a**) 13, (**b**) 20, and (**c**) 70 h, respectively.

Density values determined in the studies with pycnometer allowed us to define the ratio of the absolute volume of the alloy to the apparent (calculated) density.

Apparent density equals the ratio of mass (m) divided by the volume value (V) of a sample. This density, unlike absolute, does not include the pores contained in the material.

The value of porosity of each sintered Mg-Zn-Ca-Pr sample was measured based on the following Equation (1):

$$P = \left(1 - \frac{\rho_c}{\rho_a}\right) 100\%; \tag{1}$$

where $\rho_c$ and $\rho_a$ are the calculated value of density and the absolute density of the sample.

In order to determine the mechanical properties, the microhardness and three-point bending test were used. The hardness test was performed on the FM700 Vickers hardness tester (Future-Tech, Tokyo, Japan) with 15 s dwell time and 50 gf. The size of the indentation was measured with the aid of a calibrated microscope with a tolerance of $\pm 1/1000$ mm [52]. Multiple indentations were performed in order to obtain enough statistical data to remove invalid measurements. When measuring hardness with the Vickers method, a load of 0.49 N was used. Measurements were made by selecting the particles with the largest diameter to ensure no permanent deformation on the opposite surface. The particle diameter was

always above 1.5 d in accordance with the EN ISO 6507-1:2018-05 standard. The powders were compacted and included in resin. The mean value of the hardness of the powder mixture and sintered samples was calculated from five indentations. All of the results were presented as the mean ± standard deviation (SD).

The sintered samples with a 20 mm diameter were cut into bars and tested for their mechanical properties on a Zwick Z020 (Zwick Roell Group, Ulm, Germany) testing machine according to the EN ISO 3327 standard (the support spacing was 14 mm). Three-point bending test specimens in the form of 5 × 3 × 20 mm (width × height × length) beams were prepared from the central areas of sintered materials. The machined products were then placed in a three-point bend fixture and loaded to fracture at a rate of 2 mm/min at room temperature. A total of three specimens from each sintered sample were tested. The dimension of the specimens were not standard, but a three-point bend test was carried out for comparison purposes.

The SEM studies were performed on the $Mg_{65}Zn_{30}Ca_4Pr_1$ powder mixture, as well as on the sintered specimens after the bending test to observe the particle morphology and fracture morphology, respectively. Morphological details were investigated using the SUPRA 35 scanning electron microscope (SEM) with a voltage of 20 kV (Carl Zeiss, Jena, Germany). The studies were performed in the SE—secondary electrons mode, providing a topographic contrast. By employing the UltraDry EDS Detector (Thermo Scientific, Waltham, MA, USA), further confirmations of the qualitative chemical composition of the specimens tested in the designated areas of the material were possible. The chemical composition analysis of the powder mixture and sintered samples was performed using the Pathfinder 2.4 X-ray Microanalysis Software.

## 3. Results

### 3.1. Phase Analysis

Figure 2 shows the X-ray diffraction pattern (XRD) for the powder samples milled for 8, 13, 20, 30, and 70 h, respectively.

In the milled materials (Figure 2), the identified phases are: Mg ($P6_3/mmc$), $MgZn_2$ ($P6_3/mmc$), Zn ($P6_3/mmc$) and $Pr(OH)_3$ ($P6_3/m$). Figure 3 depicts the XRD for selected and sintered samples made from the powders after 13, 20, and 70 h of milling.

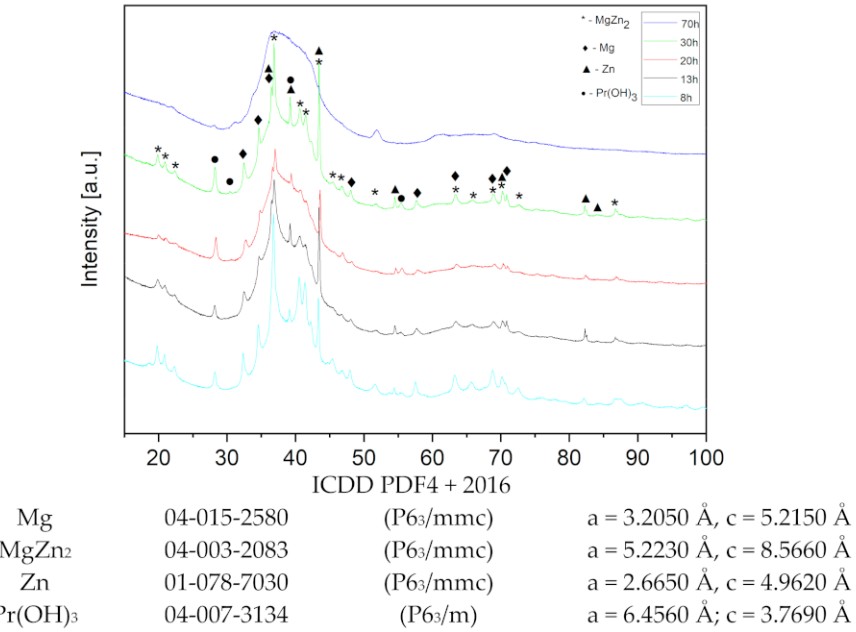

| | ICDD PDF4 + 2016 | | |
|---|---|---|---|
| Mg | 04-015-2580 | ($P6_3/mmc$) | a = 3.2050 Å, c = 5.2150 Å |
| $MgZn_2$ | 04-003-2083 | ($P6_3/mmc$) | a = 5.2230 Å, c = 8.5660 Å |
| Zn | 01-078-7030 | ($P6_3/mmc$) | a = 2.6650 Å, c = 4.9620 Å |
| $Pr(OH)_3$ | 04-007-3134 | ($P6_3/m$) | a = 6.4560 Å; c = 3.7690 Å |

**Figure 2.** Phase analysis of the Mg-Zn-Ca-Pr alloy powders milled for 8, 13, 20, 30, and 70 h.

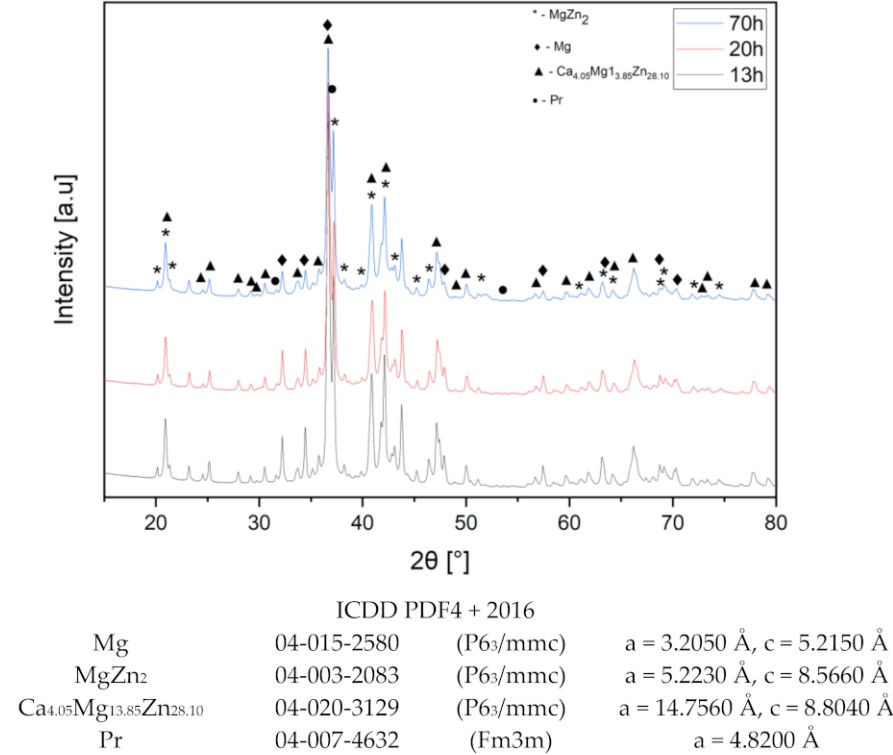

ICDD PDF4 + 2016

| | | | |
|---|---|---|---|
| Mg | 04-015-2580 | (P6$_3$/mmc) | a = 3.2050 Å, c = 5.2150 Å |
| MgZn$_2$ | 04-003-2083 | (P6$_3$/mmc) | a = 5.2230 Å, c = 8.5660 Å |
| Ca$_{4.05}$Mg$_{13.85}$Zn$_{28.10}$ | 04-020-3129 | (P6$_3$/mmc) | a = 14.7560 Å, c = 8.8040 Å |
| Pr | 04-007-4632 | (Fm3m) | a = 4.8200 Å |

**Figure 3.** Phase analysis of the selected Mg-Zn-Ca-Pr alloy samples (milled for 13, 20 and 70 h) sintered from milled powders.

In the materials milled for 70 h and sintered, the following phases were identified (Figure 3): Solution based on Mg (P6$_3$/mmc), MgZn$_2$ (P6$_3$/mmc), Ca$_{4.05}$Mg$_{13.85}$Zn$_{28.10}$ (P6$_3$/mmc) and Pr (Fm3m).

All three samples, milled at different times (13, 20 and 70 h), resulted in sintering the same final products. Slight differences can be observed only in the sizes of the crystallites of the obtained phases (Table 1). The sample, pre-milled for 70 h, as the most amorphous, resulted in sintering slightly different products with smaller crystallites (enlarged graphs—Figure 4), which could ultimately lead to obtaining samples with a higher porosity.

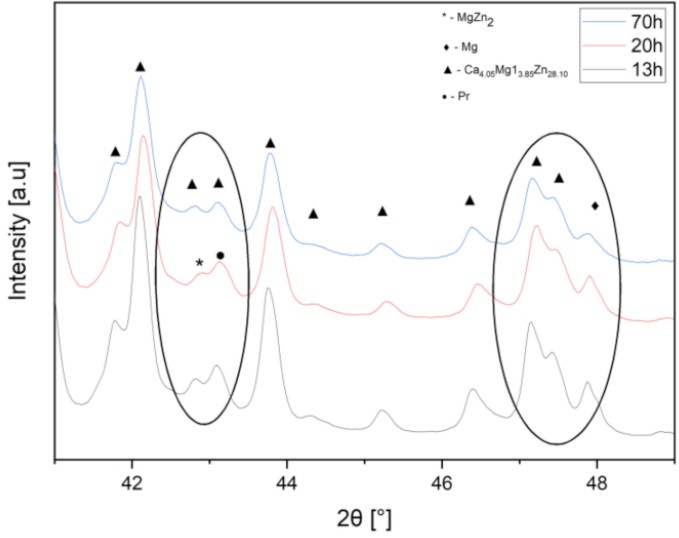

**Figure 4.** Magnification of the selected region (41–49° 2θ) of the phase analysis of the Mg-Zn-Ca-Pr alloy samples sintered from milled (70 h) powders.

**Table 1.** Crystallite sizes and changes of unit cell parameters of the main phases present in sintered Mg-Zn-Ca-Pr alloys.

| | **Mg** | | | | **MgZn₂** | | | |
|---|---|---|---|---|---|---|---|---|
| **Sample** | **Theoretical (ICDD PDF4 + Card: 04-015-2580)** | **Refined (RR) a/c [Å]** | **Crystallite Size D [Å]** | **Lattice Strain η [%]** | **Theoretical (ICDD PDF4 + Card: 04-003-2083)** | **Refined (RR) a/c [Å]** | **Crystallite Size D [Å]** | **Lattice Strain η [%]** |
| SPS_13 h | a = 3.2050 Å c = 5.2150 Å Space Group: P6₃/mmc Crystallographic System: Hexagonal | a = 3.2050 (1) c = 5.2030 (5) | 397 | 0.08 | a = 5.2230 Å c = 8.5660 Å Space Group: P6₃/mmc Crystallographic System: Hexagonal | 5.0817 (8) 9.0417 (8) | 28 | 1.17 |
| SPS_20 h | | a = 3.2042 (2) c = 5.2019(2) | 376 | 0.08 | | 5.1030 (6) 9.0108 (1) | 28 | 1.17 |
| SPS_70 h | | a = 3.2055 (7) c = 5.2070 (4) | 269 | 0.12 | | 5.0187 (2) 8.8822 (7) | 76 | 0.43 |
| | **Ca₄.₀₅Mg₁₃.₈₅Zn₂₈.₁₀** | | | | **Pr** | | | |
| Sample | Theoretical (ICDD PDF4+ card: 04-020-3129) | Refined (RR) a/c [Å] | Crystallite size D [Å] | Lattice strain η [%] | Theoretical (ICDD PDF4+ card: 04-006-4623) | Refined (RR) a/c [Å] | Crystallite size D [Å] | Lattice strain η [%] |
| SPS_13 h | a = 14.7560 Å c = 8.8040 Å; Space Group: P6₃/mmc Crystallographic System: Hexagonal | 14.7293 (2) 8.8086 (3) | 461 | 0.07 | a = 4.8200 Å Space Group: F m3m Crystallographic System: Cubic | 4.2458 (6) | 37 | 0.87 |
| SPS_20 h | | 14.7166 (8) 8.7976 (9) | 409 | 0.08 | | 4.2425 (8) | 36 | 0.90 |
| SPS_70 h | | 14.7278 (1) 8.8141 (2) | 307 | 0.16 | | 4.2020 (1) | 29 | 1.12 |

According to the Williamson–Hall theory [47], the size of the crystallites of the products in the sintered alloy after preliminary grinding for 70 h, is smaller by about 100 Å compared to the size of the crystallites of the alloys milled for 13 and 20 h (Table 1), as determined from the widening of the diffraction lines. Only the MgZn₂ phase (Table 1) shows a slightly larger crystallite size in the alloy after 70 h of grinding (28 and 76 Å, respectively).

The finely refined praseodymium (crystallite order: 30–37 Å, Table 1) indicates a much smaller unit cell compared to the reference value (difference of about 12%; ICDD PDF4 2016 base), which may indicate defects in the form of unfilled nodes and free volumes in the structure.

The remaining phases indicate slight changes in the values of the lattice constants and do not exceed 3%.

### 3.2. The Particle Size Distribution (Granulometry)

Figure 5 represents the data obtained from the granulometry test. The average particle size is shown in Table 2. The average size values for samples after 8, 13, 20, 30 and 70 h are 30, 35, 28, 20, and 17 μm, respectively.

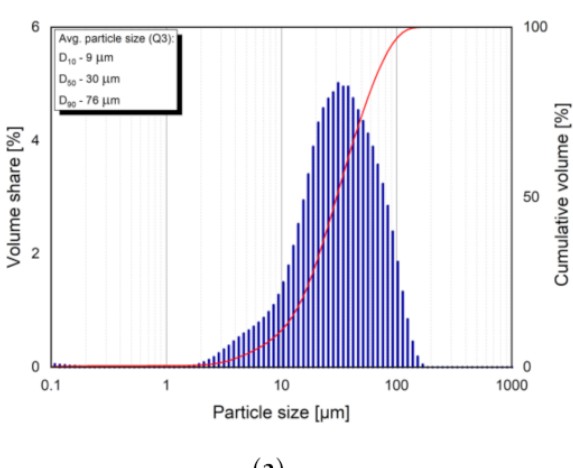

(**a**)    (**b**)

**Figure 5.** *Cont.*

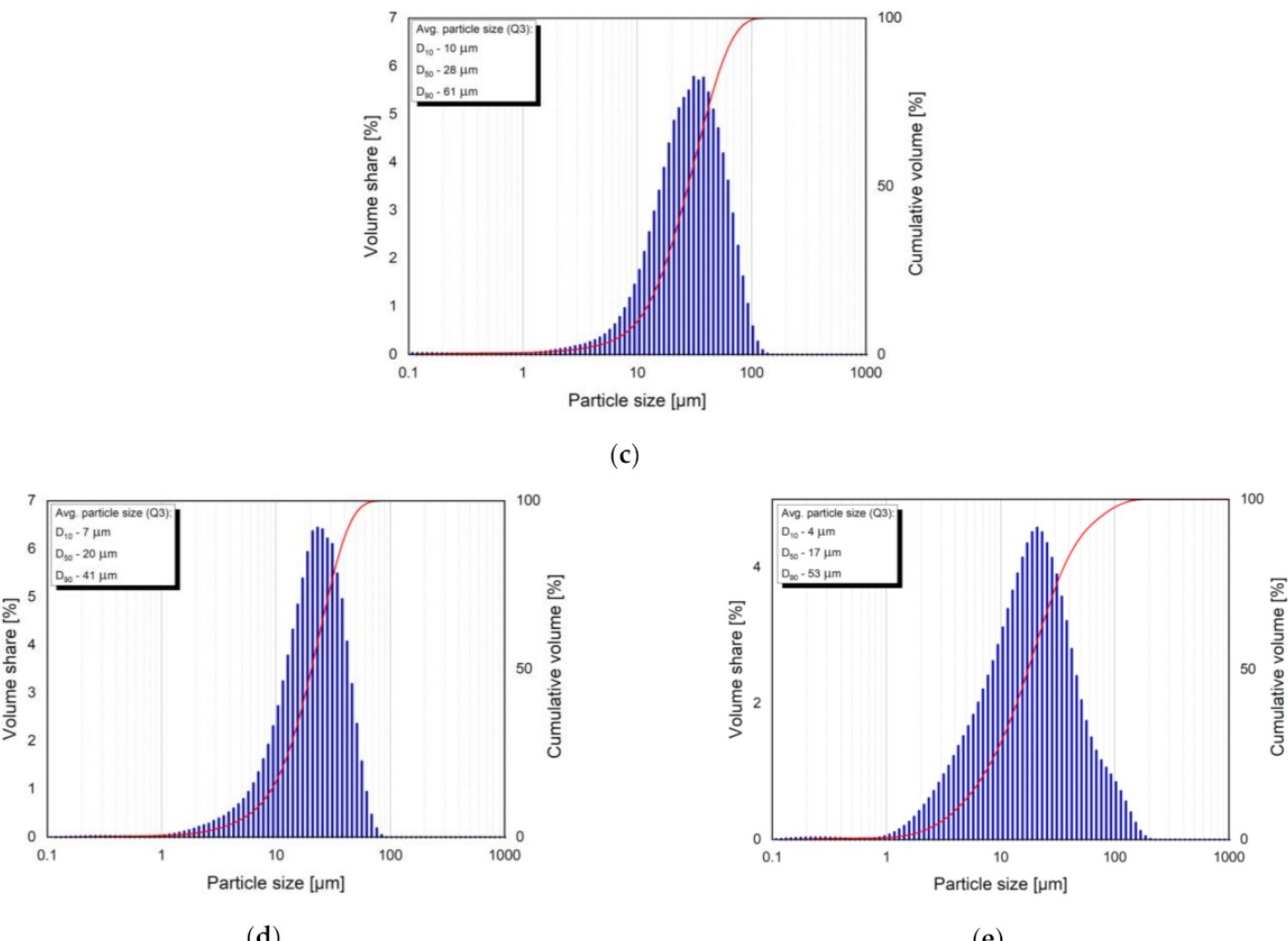

**Figure 5.** Granulometry graphs of Mg-Zn-Ca-Pr powders milled for (**a**) 8, (**b**) 13, (**c**) 20, (**d**) 30 and (**e**) 70 h, respectively. Featuring volume share (histogram) and cumulative volume (curve).

**Table 2.** Average particle size for Mg-Zn-Ca-Pr alloy powders milled for a varied amount of time.

| Avg. Particle Size [μm] | Milling Time (h) | | | | |
|---|---|---|---|---|---|
| | **8** | **13** | **20** | **30** | **70** |
| $D_{10}$ | 9 | 13 | 10 | 7 | 4 |
| $D_{50}$ | 30 | 35 | 28 | 20 | 17 |
| $D_{90}$ | 76 | 74 | 61 | 41 | 53 |

### 3.3. Density and Porosity

The average results of the powder density—$\rho_p$, the apparent density—$\rho_a$ and calculated density—$\rho_c$, as well as porosity—$P$ are presented in Table 3. The porosity varies from 0.2% for the 13 h sample, 1.0% for the 20 h sample, and 3.1% for the 70h sample. For the 13 and 20 h samples, the porosity is negligible which is in line with data presented in Figure 1, where the plateau of the sintering curves is clearly seen. It means that sintering is completed under certain conditions. For the further optimization of the SPS process of the 70h sample, the compaction pressure may be increased to enhance the densification to allow us to reduce the porosity.

**Table 3.** Average results of the powder density—$\rho_p$, the calculated density—$\rho_c$, apparent density—$\rho_a$ and porosity—$P$ for selected samples of the sintered Mg-Zn-Ca-Pr alloy for samples milled for 13, 20, and 70 h.

| Sample | Avg. $\rho_p$ (g/cm³) | Avg. $\rho_a$ (g/cm³) | $\rho_c$ (g/cm³) | $P$ (%) |
|---|---|---|---|---|
| 13 h | 3.12092 | 3.14032 | 3.133179 | 0.2% |
| 20 h | 3.13000 | 3.14606 | 3.115625 | 1.0% |
| 70 h | 3.53128 | 3.30382 | 3.200363 | 3.1% |

*3.4. Mechanical Properties*

The results of the microhardness test for the Mg-Zn-Ca-Pr alloy are gathered in Table 4. The average results of hardness amounted to 232, 262, 309, 378, and 372 $HV_{0.05}$ for samples milled for 8, 13, 20, 30, and 70 h, respectively.

Table 5 presents the microhardness values of selected sintered specimens after 13, 20, and 70 h of milling (321, 347, and 468 $HV_{0.05}$) as well as their bending strength values—$\sigma_f$. Moreover, the experimentally calculated porosity is calculated, with the lowest value of 0.2% and the highest of 3.1%. Figure 6 depicts the bending strength values obtained from the three-point bending test for selected specimens. For clarity, Young's modulus is presented in the graph in Figure 7. The bending strength values $\sigma_f$ equaled 193 MPa for the 13 h sample, 164 MPa for the 20 h sample, and 123 MPa for the 70h sample (Table 5, Figure 6).

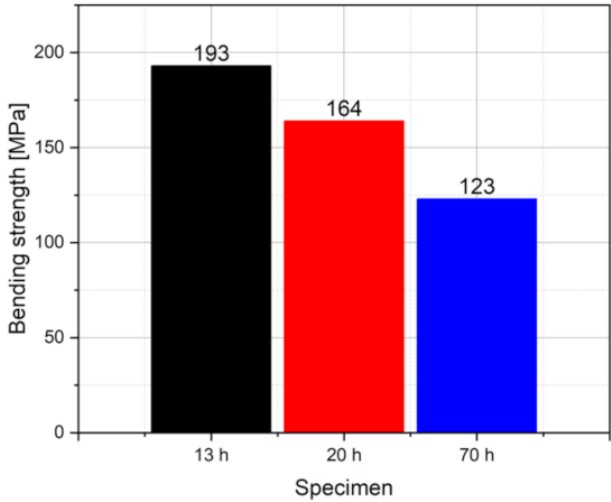

**Figure 6.** Bending strength values achieved from the three-point bending test for specimens after 13, 20, and 70 h of milling time.

**Table 4.** Microhardness results for Mg-Zn-Ca-Pr alloy powders milled for a varied amount of time. Microhardness results are presented as the mean ± standard deviation (SD).

| Milling Time (h) | Hardness Results ($HV_{0.05}$) | | | | | |
|---|---|---|---|---|---|---|
| | 1 | 2 | 3 | 4 | 5 | Avg. |
| 8 | 147 | 271 | 274 | 251 | 217 | 232 ± 53 |
| 13 | 309 | 238 | 235 | 300 | 229 | 262 ± 39 |
| 20 | 309 | 268 | 333 | 303 | 336 | 310 ± 27 |
| 30 | 394 | 329 | 364 | 419 | 385 | 378 ± 34 |
| 70 | 372 | 309 | 367 | 376 | 440 | 373 ± 46 |

**Table 5.** Microhardness and bending strength values $\sigma_f$ for Mg-Zn-Ca-Pr alloy specimens sintered from powders milled for a varied amount of time. All results are presented as the mean $\pm$ standard deviation (SD).

| Specimen | Average Hardness ($HV_{0.05}$) | $\sigma_f$ (MPa) |
|---|---|---|
| 13 h | $321 \pm 30$ | $193 \pm 7$ |
| 20 h | $347 \pm 29$ | $164 \pm 10$ |
| 70 h | $468 \pm 27$ | $123 \pm 5$ |

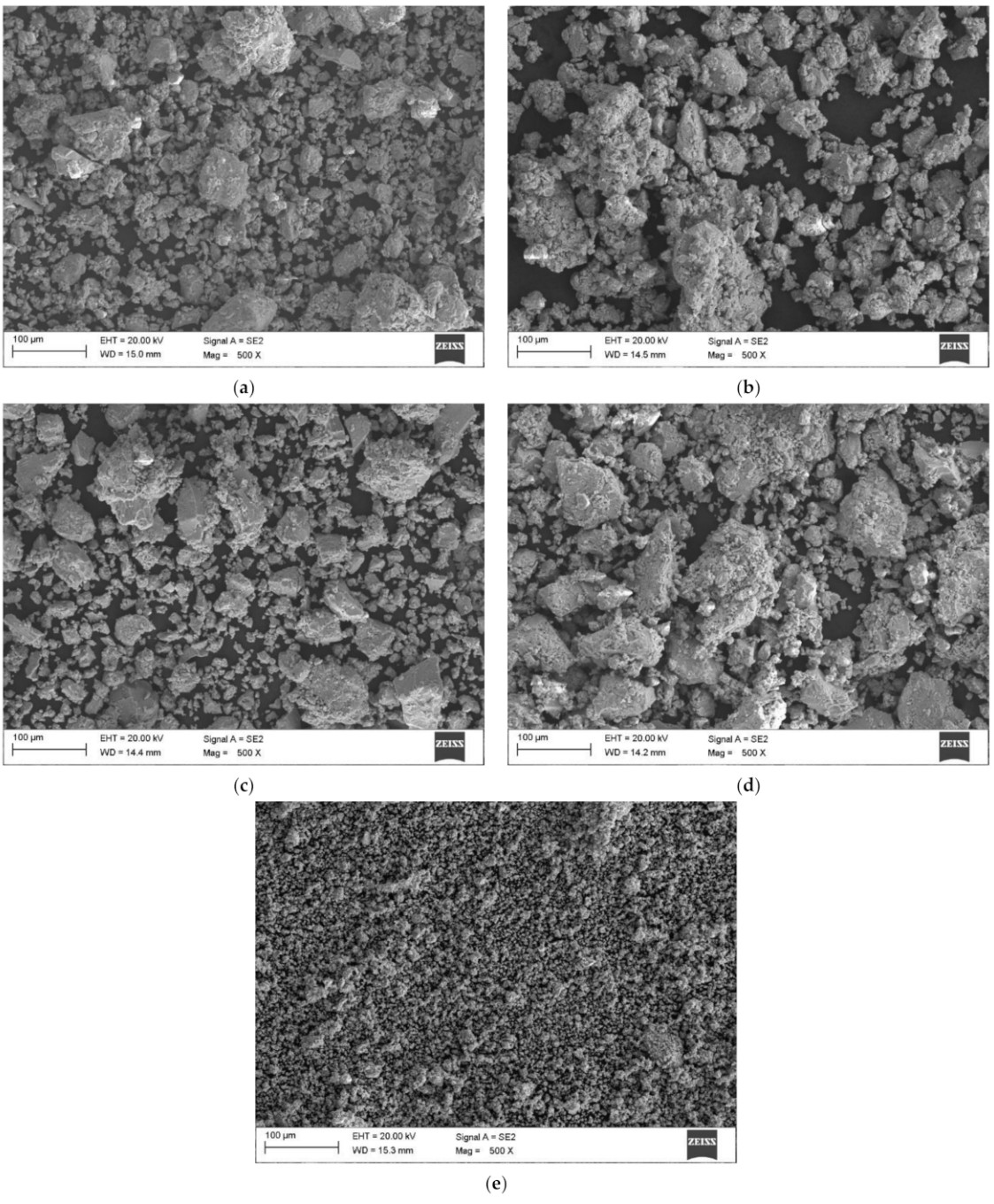

**Figure 7.** Scanning electron microscope micrographs of Mg-Zn-Ca-Pr powders milled with EDS (energy dispersive spectroscopy), results for (**a**) 8, (**b**) 13, (**c**) 20, (**d**) 30 and (**e**) 70 h, respectively.

### 3.5. Scanning Electron Microscopy

The micrographs of the powder after 8, 13, 20, 30, and 70 h of milling are presented in Figure 7a–e.

The differences in powder morphology reflecting the particle size featured in Figure 5 can be seen. Figure 8 shows the fractures of selected specimens after the three-point bending test. The cracks visible in Figure 8 bear resemblance to both trans- and intercrystalline fracture. This might have been caused by the oxide layer present on the powder before compacting. This layer creates a barrier of sorts, preventing adequate diffusion bonding at lower temperatures [53].

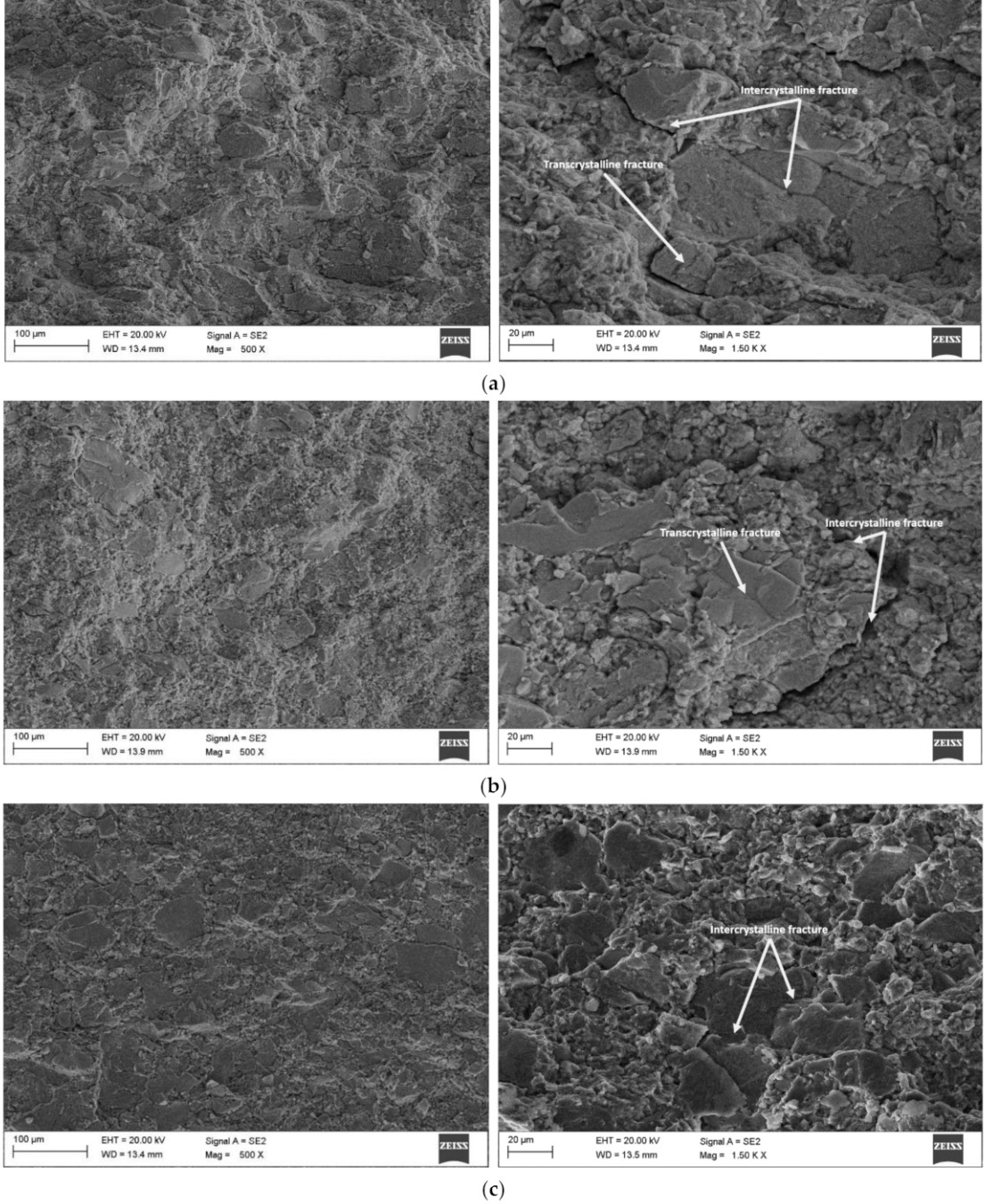

**Figure 8.** Scanning electron microscope micrographs of specimens of the Mg-Zn-Ca-Pr alloy after the three-point bending test for samples milled for (**a**) 13, (**b**) 20, and (**c**) 70 h, respectively.

Table 6 shows the results obtained via energy dispersive spectroscopy (EDS) for the powder samples, along with the intentional base chemical composition. Table 7 shows the EDS results for selected sintered samples, which were assessed based on the microstructure presented in Figure 9. Element maps which show the spatial distribution of elements in a sample after 20 h are presented in Figure 10. The areas of the components of the alloy are presented in different colors, i.e., blue, yellow, green, and purple for magnesium, zinc, calcium, and praseodymium, respectively (Figure 10). With the decrease of element concentration, the color is darker. The elements are evenly distributed in the sintered sample, with a small cluster of higher concentrations. In the area of EDS analysis (Figure 10), unreacted praseodymium is visible.

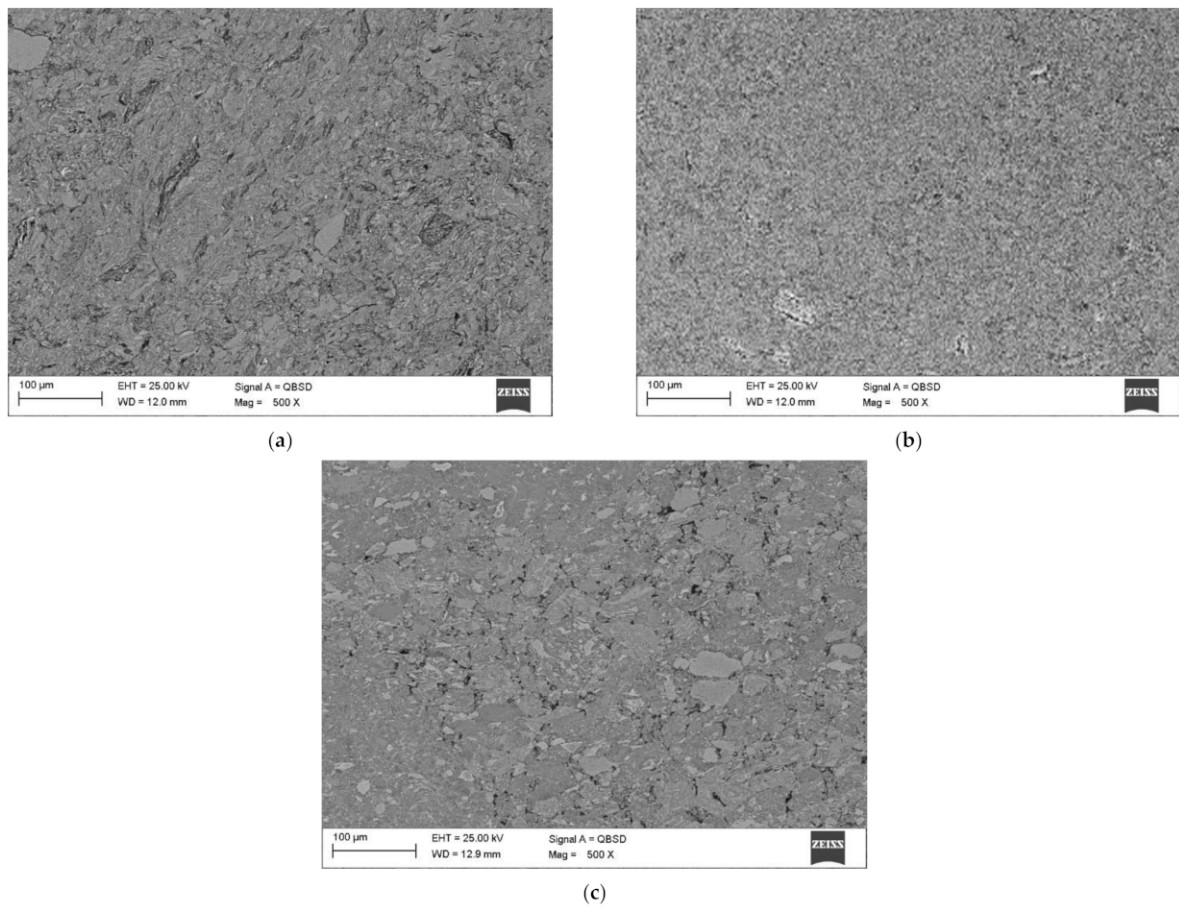

**Figure 9.** Morphology of the polished sintered specimens for the Mg-Zn-Ca-Pr alloy milled for (**a**) 13, (**b**) 20, and (**c**) 70 h.

**Table 6.** EDS results from the Mg-Zn-Ca-Pr powder alloy samples milled for 8, 13, 20, 30, and 70 h, respectively.

| Sample (Milling Time) | (wt.%) | | | | (at.%) | | | |
|---|---|---|---|---|---|---|---|---|
| | **Mg** | **Zn** | **Ca** | **Pr** | **Mg** | **Zn** | **Ca** | **Pr** |
| 8 | $36.0 \pm 1.8$ | $58.0 \pm 2.9$ | $3.0 \pm 0.2$ | $3.0 \pm 0.1$ | $60.0 \pm 3.0$ | $36.0 \pm 1.8$ | $3.0 \pm 0.2$ | 1.0 |
| 13 | $39.0 \pm 1.9$ | $55.0 \pm 2.8$ | $3.0 \pm 0.2$ | $3.0 \pm 0.1$ | $63.0 \pm 3.1$ | $33.0 \pm 1.7$ | $3.0 \pm 0.2$ | 1.0 |
| 20 | $38.0 \pm 1.9$ | $56.0 \pm 2.8$ | $4.0 \pm 0.2$ | $2.0 \pm 0.1$ | $62.0 \pm 3.1$ | $34.0 \pm 1.7$ | $4.0 \pm 0.2$ | 1.0 |
| 30 | $39.0 \pm 1.9$ | $55.0 \pm 2.7$ | $4.0 \pm 0.2$ | $3.0 \pm 0.1$ | $62.0 \pm 3.1$ | $33.0 \pm 1.7$ | $4.0 \pm 0.2$ | 1.0 |
| 70 | $36.0 \pm 1.8$ | $57.0 \pm 2.8$ | $4.0 \pm 0.2$ | $3.0 \pm 0.2$ | $61.0 \pm 3.0$ | $35.0 \pm 1.8$ | $4.0 \pm 0.2$ | 1.0 |
| Theoretical value | 41.1 | 51.0 | 4.2 | 3.7 | 65.0 | 35.0 | 4.0 | 1.0 |

**Table 7.** EDS (energy dispersive spectroscopy) results from the selected Mg-Zn-Ca-Pr alloy sintered samples milled for 13, 20, and 70 h, respectively.

| Sample (Milling Time) | (wt.%) | | | | (at.%) | | | |
|---|---|---|---|---|---|---|---|---|
| | **Mg** | **Zn** | **Ca** | **Pr** | **Mg** | **Zn** | **Ca** | **Pr** |
| 13 | 44.0 ± 2.2 | 49.0 ± 2.5 | 5.0 ± 0.2 | 2.0 ± 0.1 | 67.0 ± 3.4 | 28.0 ± 1.4 | 4.0 ± 0.2 | 1.0 |
| 20 | 41.0 ± 2.0 | 52.0 ± 2.6 | 4.0 ± 0.2 | 3.0 ± 0.1 | 64.0 ± 3.2 | 31.0 ± 1.5 | 4.0 ± 0.2 | 1.0 |
| 70 | 44.0 ± 2.2 | 49.0 ± 2.5 | 4.0 ± 0.2 | 3.0 ± 0.2 | 67.0 ± 3.4 | 28.0 ± 1.4 | 4.0 ± 0.2 | 1.0 |
| **Theoretical value** | 41.1 | 51.0 | 4.2 | 3.7 | 60.0 | 35.0 | 4.0 | 1.0 |

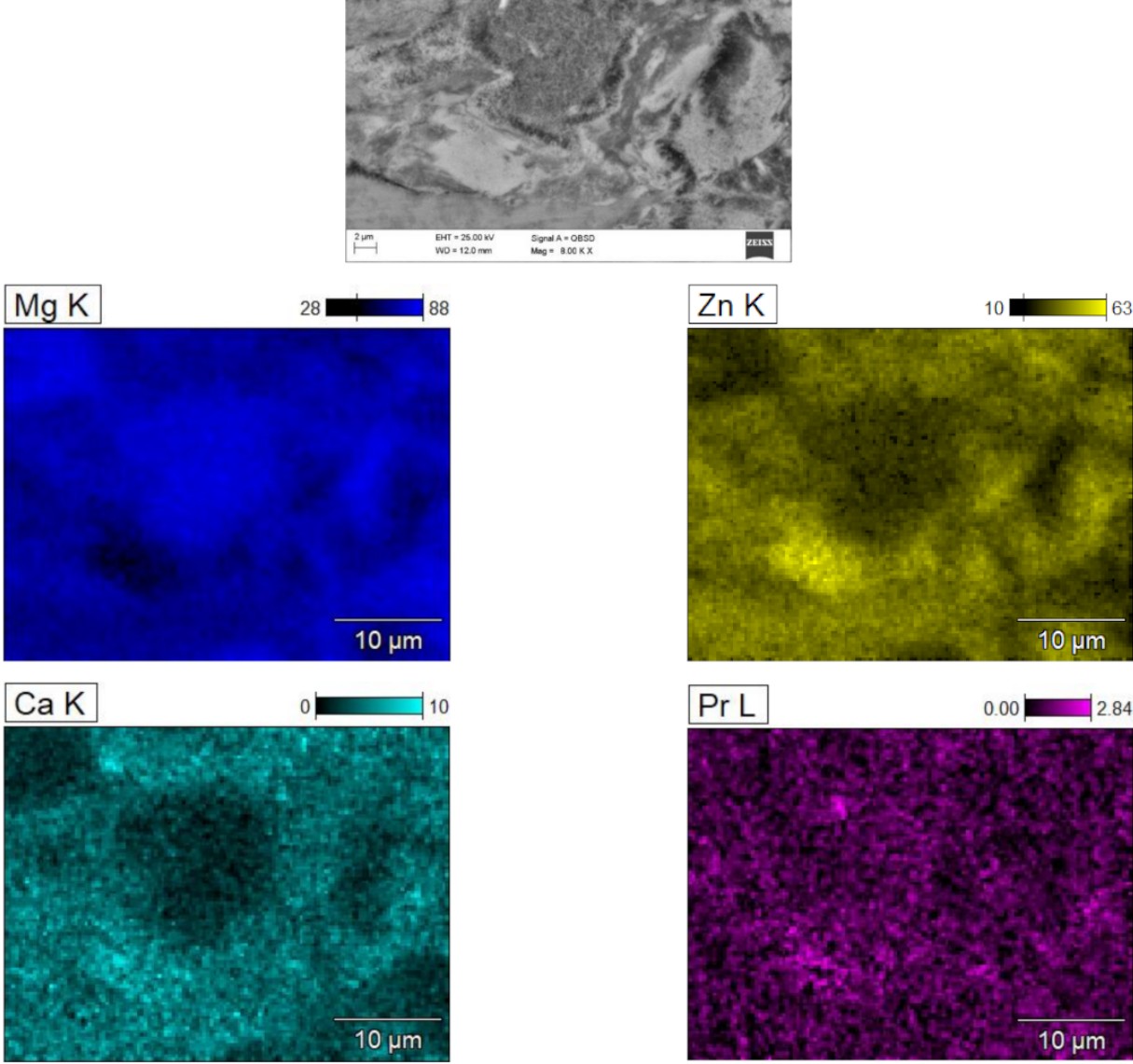

**Figure 10.** Element maps show the spatial distribution of elements performed on the energy dispersive spectrometer for the selected sample after 20 h of milling time. The areas of the components of the alloy are presented in different colors, i.e., blue, yellow, green, and purple for magnesium, zinc, calcium, and praseodymium, respectively.

## 4. Discussion

The XRD patterns shown in Figure 2 present phases formed during the milling stage of the experiment. The wide peak between 35° and 45° indicates that the grinding products are partially amorphous, with visible peaks of microcrystalline phases, mainly, pure Mg and Zn, and $MgZn_2$ phase. The presence of $MgZn_2$ may be beneficial to the alloy, as it is very stable, both thermally and mechanically, and usually is a desirable Laves phase, due to its strengthening and stabilization factor of the alloy [54].

The microhardness results presented in Table 4 show a rising tendency from 232 $HV_{0.05}$ after 8 h of milling to 378 $HV_{0.05}$ after 30 h of milling. The result after 70 h averages around 372 $HV_{0.05}$, yet the differences between the results for 30 h and 70 h are negligible (Table 4). Those results can be correlated to the granulometry results presented in Figure 5 and Table 2, where the average particle size is presented. The average particle size after 8 h equals 30 μm and rises to 35 μm after 13 h (Table 2). This is due to the consolidation of the finer particles into bigger agglomerates during the milling. The resulting microstructure can be observed in Figure 7b, where smaller particles are embedded into bigger ones. Such a structure is not visible in Figure 7a. After 13 h the particles decrease sharply along with the milling time, upon reaching 20 μm after 30 h. Prolonged milling at this stage does not decrease the particle size a considerable amount. The particle size (Table 2) can be correlated to the microhardness (Table 4), which changes in a similar manner. While the hardness rises, the average particle size decreases until they reach stability.

The micrographs of the powder samples featured in Figure 7 show how the structure of the powder changes along with the milling time. Fine particulate agglomerates into bigger particles, to be crushed again as the cycle repeats. It is worth noting that the structure in Figure 7b,d resembles the "corncob" structure mentioned by Jurczyk [43], meaning all the base materials are brittle. The plate structure usually seen in mechanical synthesis is not present in this case, as for the brittle materials, a very strong fragmentation of the samples causes a sharp rise in the hardness of the finer particulate leading to trapping them in the crevices of bigger, softer agglomerates, thus breaking them apart in the process. Moreover, the EDS results (Table 6) show the chemical composition of the powder alloys, which does not change drastically between the samples, meaning it is homogeneously distributed during the milling. It may have a very vital effect on the corrosion resistance of the alloy, as due to the homogeneous distribution the probability of galvanic corrosion between different regions of the sample is drastically reduced.

After sintering, the samples were studied again. The plateau visible in Figures 3 and 4 proves that the densification process was completed for sintering with used parameters [40,55]. The elements are evenly distributed in the sintered sample (Table 7), with a small cluster of higher concentrations (Figure 10, Table 1). A considerable rise in hardness can be observed (Tables 4 and 5, Figure 11). From the selected samples the hardness of the sample rose from 262 to 321 $HV_{0.05}$ after 13 h, 309 to 347 $HV_{0.05}$ after 20 h, and from 372 to 468 $HV_{0.05}$ after 70 h. This change is directly influenced by the compacting pressure during the sintering process, as the gaps between the powder particles were compacted resulting in negligible porosity as reported in Table 3 and Figure 9. The sinter hardness increase is relatable to the powder hardness increase presented in Table 3. The sharp hardness increase of the sample after 70 h milling time is caused by the increase of the strain in the powder particles, which grows alongside the milling time. Although the sample is much harder than its counterparts after 13 and 20 h, its bending strength is much lower due to its higher porosity (3.13%) (Tables 3 and 5, Figures 6 and 9) [56].

The porosity of the sintered sample manufactured from the powder after 70 h of milling (Figure 9, Table 3), is higher than the porosities of the samples milled for 13 and 20 h due to the presence of the amorphous phase (Figure 4). The free volume present in amorphous materials did not allow us to obtain a lower porosity (Table 3), albeit being sintered with the same parameters (Table 5) [57]. Thus, the results of the bending strength values $\sigma_f$ obtained via the three-point bending test achieved values of 193 MPa for the 13 h sample, 164 MPa for the 20 h sample, and 123 MPa for the 70 h sample (Figure 6, Table 5).

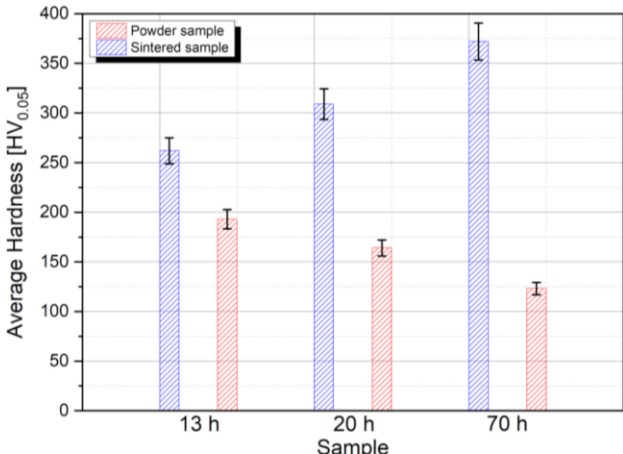

**Figure 11.** Average hardness and particle size for Mg-Zn-Ca-Pr samples after 13, 20, and 70 h of milling, for both powder and sintered samples.

The value of 164 MPa (Figure 6, Table 5) is very close to the bending resistance of the human cortical bone, which is a very important fact as the materials for implants should have very similar mechanical properties to the bone they are supporting. This relation is best described by Wollf's law, stating that the trabeculae may reflect the loading on the bone performing adaptive changes followed by secondary changes to the external part of the bone [13,58–60]. The bone adapts to the load by increasing its density, this is true for the opposite as well. After introducing an implant capable of bearing a much higher load, the bone may reduce its density due to the lack of stimulus for its reconstruction. Such a reduction of bone density is known as osteopenia. It is a pathological state and an early step of osteoporosis.

Figure 8 shows the SEM micrographs of the specimens after fracture, and features brittle fracture for all the specimens, although Figure 8c is slightly more plastic as compared to others. As previously mentioned, no porosity or porosity of negligible impact can be seen in the fractures. Slight cracks can be observed on the grain boundary in Figure 8a,c.

## 5. Conclusions

The major findings of this study can be summarized as follows:

- The HEMA method was applied to synthesize Mg-Zn-Ca-Pr powders and SPS to consolidate the $Mg_{65}Zn_{30}Ca_4Pr_1$ alloy.
- HEMA is an effective method for preparing Mg-Zn-Ca-Pr powders. Powders of different grain sizes were successfully synthesized by mechanically milling powders which consist of Mg phase and fine particles of Pr for 8–70 h.
- The microstructure evolution and morphology of the as-milled powders were observed; after milling for 13, 20 and 70 h, the crystallite size of the solution based on Mg structure in Mg-Zn-Ca-Pr powders was refined to 397, 376 and 269 Å, respectively. The average size of the Pr particles in Mg-Zn-Ca-Pr powders was refined to about 30 Å for all these samples.
- A superior combination of mechanical properties was attained after sintering powder after 20 h of milling time. Hardness increased significantly with increasing the milling time up to 70 h, which is attributed to both grain refinement and the formation of secondary phases (solution based on Mg ($P6_3/mmc$), $MgZn_2$ ($P6_3/mmc$), $Ca_{4.05}Mg_{13.85}Zn_{28.10}$ ($P6_3/mmc$) and Pr (Fm3m)). During the HEMA and SPS processes, the Young's modulus value decreased sharply for milling times of 70 h. This decrease was associated with the increase in the porosity observed in the sintered samples.
- The effect of holding time was insignificant and sufficient compaction was achieved with a 4 min holding time, which could prospectively lead to a reduction in production costs.

**Author Contributions:** Conceptualization, S.L. and B.H.; methodology, S.L., B.H. and J.P.; validation, K.G., S.L., M.K. (Małgorzata Karolus); investigation, B.H., S.L., J.P., M.K. (Małgorzata Karolus), M.K. (Marek Kremzer), K.G., J.W., R.R. and D.G.; resources, S.L., D.G.; writing—original draft, B.H.; writing—review and editing, B.H., S.L. and M.K. (Małgorzata Karolus); supervision, S.L., M.K. (Małgorzata Karolus) and D.G. All authors have read and agreed to the published version of the manuscript.

**Funding:** This research was funded by the National Science Center, Poland, grant number 2017/27/B/ST8/02927.

**Institutional Review Board Statement:** Not applicable.

**Informed Consent Statement:** Not applicable.

**Data Availability Statement:** Data sharing is not applicable to this article.

**Conflicts of Interest:** The authors declare no conflict of interest.

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
