# Peer review of "Microstructure and Mechanical Properties of Spark Plasma Sintered Mg-Zn-Ca-Pr Alloy"

_metals, doi:10.3390/met12030375_

Round 1
Reviewer 1 Report
In this publication the authors investigate the effect of High Energy Mechanical Alloying (HEMA) on powder properties and the properties of bulk Mg-Zn-Ca-Pr specimen consolidated by spark plasma sintering. The motivation for this investigation is understandable and can be found in the introduction.
Nevertheless, in the publication of S. Lesz et. al: “Characteristics of the Mg-Zn-Ca-Gd Alloy after Mechanical Alloying, DOI: 10.3390/ma14010226, the powder properties of a comparable magnesium alloy were investigated.
Q1: Which influence of the mechanical alloying on the powder properties of the magnesium alloy is expected by the use of 1 at.% praseodymium instead of 1 at.% gadolinium? The authors need to clearly indicate what´s the novelty of this publication in addition to the investigation of the bulk properties of the sintered samples.
Q2: Introduction/second paragraph: “Magnesium and its alloys have… …, and good corrosion resistance.” Does this statement refer to the specific application? In general, this statement is incorrect.
Q3: Materials and Methods/first paragraph: For the terms in parentheses, the formulation: “purity of 99.99 %” is more common.
Q4: What was the diameter and the material of the grinding balls?
Q5: Regarding the sintering process some important information are missing: dwell time at maximum temperature, heating rate, time of application of the pressure (50 MPa), sample size (the sample diameter can only be found in the explanations for the bending test). Furthermore, Figure 1 was not discussed: Why do the authors show the punch displacement? How can the authors explain the temperature curve at the time of about 7 min and 30 s – is this a material or temperature control effect? The caption of the figure should be corrected (heating is not the correct term).
Q6: How was the measurement of the microhardness on the powder samples carried out? The length of the indentation diagonals is in the range of the particle size – how was the correct hardness measurement ensured?
Q7: Regarding the three-point bending tests all important information are missing: dimensions of the test bars, calculation of the bending modulus (Is it ensured that the bending modulus corresponds to the elastic modulus?), test execution (What was measured – load and deflection?).
Q8: What was the reason that the authors decided to separate the results and the discussion? For the reader it is difficult to understand the context in this way. In any case, the results of the investigation of the powder properties should by separated from the results of the investigation of the sintered sample properties.
Q9: The quality of some figures needs to be improved - In any case the quality of Figure 4.
Q10: Table 2: The terms “10%”, “20%” and “90%” should be substituted by the d10, d50 and d90, respectively.
Q11: Table 4/5, Figure 7: The authors should provide the mean value and the standard deviation. Furthermore, Figure 7 should be removed as all of the information in this figure are contained in Table 5.
Q12: Figure 6: The stress strain curves in this figure show an unusual trend and should be corrected. How can the authors explain the material behavior (the different slope) during the three-point bending test? Thus, the values of the elasticity modulus are questionable (please, see Q7).
Q13: Line 256-257: “Figure 9 shows the fractures of selected samples after the three-point bending test. The cracks visible in Figure 9 bear resemblance to both trans- and intercrystalline fracture.”
From which sample area were the samples for the SEM investigation taken? Furthermore, please mark the cracks in the corresponding micrographs.
Q14: Based on the above annotations, the discussion and conclusions section of the publication need to improved.
Reviewer 2 Report
This paper investigated the effect of the milling time on density, microstructure, phase composition, and mechanical properties of Mg-Zn-Ca-Pr powders processed by High Energy Mechanical Alloying (HEMA) and consolidated by spark plasma sintering (SPS). The paper is interesting as it uses combination of mechanical alloying and SPS in synthesizing an Mg-based alloy. Moreover, the manuscript is rich in experimental techniques, including SEM, XRD, and EDX. Although it is a very interesting read, it needs some major revisions before it can be published. Here is the list of comments:
- The English needs to be polished. Some sentences are rather long, and sometimes it is difficult to follow the text
- The introduction to the SPS is very poor. Only a few lines are devoted to introduce the technique and to the mechanisms of consolidation during SPS. I strongly recommend authors have a look at the following papers and use discussions in this paper to elaborate more on the topic:
- An Investigation on the Microstructure, Interface, and Mechanical Properties of Spark Plasma Sintered Ni/Ni-Ni3Al/Ni Compound, Journal of Materials Engineering and Performance, 1-7, 2021
- Towards a High Strength Ductile Ni/Ni3Al/Ni Multilayer Composite using Spark Plasma Sintering, Science of Sintering 51 (4), 2019
- Microstructure and Mechanical Properties of Spark Plasma Sintered Nanocrystalline TiAl-xB Composites (0.0< x< 1.5 at.%) Containing Carbon Nanotubes, Journal of Materials Engineering and Performance 30 (6), 4380-4392, 2021
- The following statement in the conclusion is more a generic statement and is not really a conclusion of this investigation. I suggest authors either remove this statement or re-write it with a more specific message: “A wide range of mechanical properties can be obtained by altering the SPS parameters to achieve case-specific requirements typical of biomedical materials. Favorable properties of the Mg65Zn30Ca4Pr1 alloy together with the ability to fine-tune its performance by varying the SPS parameters, therefore, make it a particularly promising material for biomedical use.”
- Quality of images are rather poor. If possible, please replace them with better images
- Please provide more information on the number of each test and how repeatability of tests are assessed/considered (especially mechanical tests).
Reviewer 3 Report
In this work, Mg-Zn-Ca-Pr alloy were fabricated by high Energy Mechanical Alloying and spark plasma sintering (SPS) technique. The effect of the milling time on density, microstructure, phase composition, mechanical properties were fully investigated. However, there are several points should be addressed before this article is considered for publication:
- XRD curves are overlapped and need to be separated.
- Figures need to replot. The curve in Figure 1 was broken. Figure 4 was not clear.
- Some results need to double check. The young's modulus of magnesium alloy is only 1 ~ 2GPa?
- There was some mistakes of the three-point bending results.
Round 2
Reviewer 1 Report
There is a small mistake in this sentence:
Three-point bending test specimens in the form of 5×3×20 mm (width×height-length) beams were prepared from the central areas of sintered materials.
"(width x height x length)"
Reviewer 2 Report
Thanks for considering my comments. The paper is now suitable for publication.
Reviewer 3 Report
The author has revised all the questions and the revised manuscript looks much better.